# Predicting floor level for 911 Calls with Neural Networks and Smartphone Sensor Data

**William Falcon, Henning Schulzrinne**
Department of Computer Science
Columbia University
New York, NY 10027, USA
{waf2107,hgs}@columbia.edu

## Abstract

In cities with tall buildings, emergency responders need an accurate floor level location to find 911 callers quickly. We introduce a system to estimate a victim's floor level via their mobile device's sensor data in a two-step process. First, we train a neural network to determine when a smartphone enters or exits a building via GPS signal changes. Second, we use a barometer equipped smartphone to measure the change in barometric pressure from the entrance of the building to the victim's indoor location. Unlike impractical previous approaches, our system is the first that does not require the use of beacons, prior knowledge of the building infrastructure, or knowledge of user behavior. We demonstrate real-world feasibility through 63 experiments across five different tall buildings throughout New York City where our system predicted the correct floor level with $100\%$ accuracy.

## 1    Introduction

Indoor caller floor level location plays a critical role during 911 emergency calls. In one use case, it can help pinpoint heart attack victims or a child calling on behalf of an incapacitated adult. In another use case, it can help locate firefighters and other emergency personnel within a tall or burning building. In cities with tall buildings, traditional methods that rely on GPS or Wi-Fi fail to provide reliable accuracy for these situations. In these emergency situations knowing the floor level of a victim can speed up the victim search by a factor proportional to the number of floors in that building. In recent years methods that rely on smartphone sensors and Wi-Fi positioning (Xue et al., 2017) have been used to formulate solutions to this problem.

In this paper we introduce a system that delivers an estimated floor level by combining deep learning with barometric pressure data obtained from a Bosch bmp280 sensor designed for "floor level accuracy" (Bosch, 2016) and available in most smartphones today[1]. We make two contributions: the first is an LSTM (Hochreiter & Schmidhuber, 1997) trained to classify a smartphone as either indoors or outdoors (IO) using GPS, RSSI, and magnetometer sensor readings. Our model improves on a previous classifier developed by (Avinash et al., 2010). We compare the LSTM against feedforward neural networks, logistic regression, SVM, HMM and Random Forests as baselines. The second is an algorithm that uses the classifier output to measure the change in barometric pressure of the smartphone from the building entrance to the smartphone's current location within the building. The algorithm estimates the floor level by clustering height measurements through repeated building visits or a heuristic value detailed in section 4.5.

We designed our method to provide the most accurate floor level estimate without relying on external sensors placed inside the building, prior knowledge of the building, or user movement behavior. It merely relies on a smartphone equipped with GPS and barometer sensors and assumes an arbitrary

---

[1]As of June 2017 the market share of phones in the US is 44% Apple and 29.1% Samsung (ComScore, 2017). 74% are iPhone 6 or newer (Comscore, 2017). The iPhone 6 has a barometer (Apple, 2017). Models after the 6 still continue to have a barometer. For the Samsung phones, the Galaxy s5 is the most popular (Piejko, 2017), and has a barometer (Samsung, 2014)

user could use our system at a random time and place. We offer an extensive discussion of potential real-world problems and provide solutions in (appendix B).

We conducted 63 test experiments across six different buildings in New York City to show that the system can estimate the floor level inside a building with 65.0% accuracy when the floor-ceiling distance in the building is unknown. However, when repeated visit data can be collected, our simple clustering method can learn the floor-ceiling distances and improve the accuracy to 100%. All code, data and data collection app are available open-source on github.[2].

## 2    RELATED WORK

Current approaches used to identify floor level location fall into two broad categories. The first method classifies user activity, i.e., walking, stairs, elevator, and generates a prediction based on movement offset (Avinash et al., 2010) (Song et al., 2014). The second category uses a barometer to calculate altitude or relative changes between multiple barometers (Parviainen et al., 2008) (Li et al., 2013) (Xia et al., 2015). We note that elevation can also be estimated using GPS. Although GPS works well for horizontal coordinates (latitude and longitude), GPS is not accurate in urban settings with tall buildings and provides inaccurate vertical location (Lammel et al., 2009).

Song et al. (2014) describe three modules which model the mode of ascent as either elevator, stairs or escalator. Although these methods show early signs of success, they required infrastructure support and tailored tuning for each building. For example, the iOS app (Wonsang, 2014) used in this experiment requires that the user state the floor height and lobby height to generate predictions.

Avinash et al. (2010) use a classifier to detect whether the user is indoors or outdoors. Another classifier identifies whether a user is walking, climbing stairs or standing still. For the elevator problem, they build another classifier and attempt to measure the displacement via thresholding. While this method shows promise, it needs to be calibrated to the user's step size to achieve high accuracy, and the floor estimates rely on observing how long it takes a user to transition between floors. This method also relies on pre-training on a specific user.

Li et al. (2013) conduct a study of the feasibility of using barometric pressure to generate a prediction for floor level. The author's first method measures the pressure difference between a reference barometer and a "roving" barometer. The second method uses a single barometer as both the reference and rover barometer, and sets the initial pressure reading by detecting Wi-Fi points near the entrance. This method also relies on knowing beforehand the names of the Wi-Fi access points near the building entrance.

Xia et al. (2015) equip a building with reference barometers on each floor. Their method thus allows them to identify the floor level without knowing the floor height. This technique also requires fitting the building with pressure sensors beforehand.

## 3    DATA DESCRIPTION

To our knowledge, there does not exist a dataset for predicting floor heights. Thus, we built an iOS app named Sensory[3] to aggregate data from the smartphone's sensors. We installed Sensory on an iPhone 6s and set to stream data at a rate of 1 sample per second. Each datum consisted of the following: indoors, created at, session id, floor, RSSI strength, GPS latitude, GPS longitude, GPS vertical accuracy, GPS horizontal accuracy, GPS course, GPS speed, barometric relative altitude, barometric pressure, environment context, environment mean bldg floors, environment activity, city name, country name, magnet x, magnet y, magnet z, magnet total.

Each trial consisted of a continuous period of Sensory streaming. We started and ended each trial by pressing the start button and stop button on the Sensory screen. We separated data collection by two different motives: the first to train the classifier, the second to make floor level predictions. The same sensory setup was used for both with two minor adjustments: 1) Locations used to train the

---

[2]https://github.com/williamFalcon/Predicting-floor-level-for-911-Calls-with-Neural-Networks-and-Smartphone-Sensor-Data

[3]https://github.com/williamFalcon/sensory

classifier differed from locations used to make building predictions. 2) The indoor feature was only used to measure the classifier accuracy in the real world.

## 3.1 Data Collection for Indoor-Outdoor Classifier

Our system operates on a time-series of sensor data collected by an iPhone 6s. Although the iPhone has many sensors and measurements available, we only use six features as determined by forests of trees feature reduction (Kawakubo & Yoshida, 2012). Specifically, we monitor the smartphone's barometric pressure $P$, GPS vertical accuracy $GV$, GPS horizontal accuracy $GH$, GPS Speed $GS$, device RSSI[4] level $rssi$ and magnetometer total reading $M$. All these signals are gathered from the GPS transmitter, magnetometer and radio sensors embedded on the smartphone. Appendix table 5 shows an example of data points collected by our system. We calculate the total magnetic field strength from the three-axis $x, y, z$ provided by the magnetometer by using equation 1. Appendix B.5 describes the data collection procedure.

$$M = \sqrt{x^2 + y^2 + z^2} \tag{1}$$

### 3.1.1 Data Collection for Floor Prediction

The data used to predict the floor level was collected separately from the IO classifier data. We treat the floor level dataset as the testing set used only to measure system performance. We gathered 63 trials among five different buildings throughout New York City to explore the generality of our system. Our building sampling strategy attempts to diversify the locations of buildings, building sizes and building material. The buildings tested explore a wide-range of possible results because of the range of building heights found in New York City (Appendix 4). As such, our experiments are a good representation of expected performance in a real-world deployment.

The procedure described in appendix B.6 generates data used to predict a floor change from the entrance floor to the end floor. We count floor levels by setting the floor we enter to 1. This trial can also be performed by starting indoors and walking outdoors. In this case, our system would predict the person to be outdoors. If a building has multiple entrances at different floor levels, our system may not give the correct numerical floor value as one would see in an elevator. Our system will also be off in buildings that skip the 13th floor or have odd floor numbering. The GPS lag tended to be less significant when going from inside to outside which made outside predictions trivial for our system. As such, we focus our trials on the much harder outside-to-inside prediction problem.

### 3.1.2 Data Collection for Floor Clustering

To explore the feasibility and accuracy of our proposed clustering system we conducted 41 separate trials in the Uris Hall building using the same device across two different days. We picked the floors to visit through a random number generator. The only data we collected was the raw sensor data and did not tag the floor level. We wanted to estimate the floor level via entirely unsupervised data to simulate a real-world deployment of the clustering mechanism. We used both the elevator and stairs arbitrarily to move from the ground floor to the destination floor.

## 4 Methods

In this section, we present the overall system for estimating the floor level location inside a building using only readings from the smartphone sensors First, a classifier network classifies a device as either indoors or outdoors. The next parts of the algorithm identify indoor/outdoor transitions (IO), measure relative vertical displacement based on the device's barometric pressure, and estimate absolute floor level via clustering.

### 4.1 Data Format

From our overall signal sequence $\{x_1, x_2, x_j, ..., x_n\}$ we classify a set of $d$ consecutive sensor readings $X_i = \{x_1, x_2, ..., x_d\}$ as $y = 1$ if the device is indoors or $y = 0$ if outdoors. In our experiments

---

[4]As measured by the iOS status bar

we use the middle value $x_j$ of each $X_i$ as the $y$ label such that $X_i = \{x_{j-1}, x_j, x_{j+1}\}$ and $y_i = x_j$. The idea is that the network learns the relationship for the given point by looking into the past and future around that point. This design means our system has a lag of $d/2 - 1$ second before the network can give a valid prediction. We chose $d = 3$ as the number of points in $X$ by random-search (Bergstra & Bengio, 2012) over the point range $[1, 30]$. Fixing the window to a small size $d$ allows us to use other classifiers as baselines to the LSTM and helps the model perform well even over sensor reading sequences that can span multiple days.

## 4.2 Indoor-Outdoor Classification Neural Networks

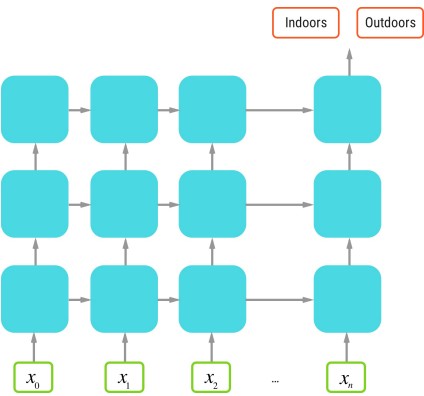

Figure 1: LSTM network architecture. A 3-layer LSTM. Inputs are sensor readings for $d$ consecutive time-steps. Target is $y = 1$ if indoors and $y = 0$ if outdoors.

The first key component of our system is a classifier which labels each device reading as either indoors or outdoors (IO). This critical step allows us to focus on a subset of the user's activity, namely the part when they are indoors. We conduct our floor predictions in this subspace only. When a user is outside, we assume they are at ground level. Hence our system does not detect floor level in open spaces and may show a user who is on the roof as being at ground level. We treat these scenarios as edge-cases and focus our attention on floor level location when indoors.

Although our baselines performed well, the neural networks outperformed on the test set. Furthermore, the LSTM serves as the basis for future work to model the full system within the LSTM; therefore, we use a 3-layer LSTM as our primary classifier. We train the LSTM to minimize the binary cross-entropy between the true indoor state $y$ of example $i$ and the LSTM predicted indoor state $\text{LSTM}(X) = \hat{y}$ of example $i$. This objective cost function $C$ can be formulated as:

$$C(y_i, \hat{y}_i) = \frac{1}{n} \sum_{i=1}^{n} -(y_i log(\hat{y}_i) + (1 - y_i) log(1 - \hat{y}_i)) \tag{2}$$

Figure 4.2 shows the overall architecture. The final output of the LSTM is a time-series $T = \{t_1, t_2, ..., t_i, t_n\}$ where each $t_i = 0, t_i = 1$ if the point is outside or inside respectively.

The IO classifier is the most critical part of our system. The accuracy of the floor predictions depends on the IO transition prediction accuracy. The classifier exploits the GPS signal, which does not cut out immediately when walking into a building. We call this the "lag effect." The lag effect hurts our system's effectiveness in 1-2 story houses, glass buildings, or ascend methods that may take the user near a glass wall.

A substantial lag means the user could have already moved between floors by the time the classifier catches up. This will throw off results by 1-2 floor levels depending on how fast the device has moved between floors. The same effect is problematic when the classifier prematurely transitions indoors as the device could have been walking up or down a sloped surface before entering the building. We correct for most of these errors by looking at a window of size $w$ around the exact classified transition point. We use the minimum device barometric pressure in this window as our $p_0$. We set $w = 20$ based on our observations of lag between the real transition and our predicted

transition during experiments. This location fix delay was also observed by (Nawarathne et al., 2014) to be between 0 and 60 seconds with most GPS fixes happening between 0 and 20 seconds.

## 4.3 INDOOR-OUTDOOR TRANSITION DETECTOR

Figure 2: To find the indoor/outdoor transitions, we convolve filters $V_1, V_2$ across timeseries of Indoor/Outdoor predictions $T$ and pick each subset $s_i$ with a Jaccard distance $\geq 0.4$. The transition $t_i$ is the middle index in set $s_i$.

Given our LSTM IO predictions, we now need to identify when a transition into or out of a building occurred. This part of the system classifies a sub-sequence $s_i = T_{i:i+|V_1|}$ of LSTM predictions as either an IO transition or not. Our classifier consists of two binary vector masks $V_1, V_2$

$$V_1 = [1, 1, 1, 1, 1, 0, 0, 0, 0, 0] \tag{3}$$

$$V_2 = [0, 0, 0, 0, 0, 1, 1, 1, 1, 1] \tag{4}$$

that are convolved across $T$ to identify each subset $s_i \in S$ at which an IO transition occurred. Each subset $s_i$ is a binary vector of in/out predictions. We use the Jaccard similarity (Choi et al., 2010) as an appropriate measure of distance between $V_1, V_2$ and any $s_i$.

As we traverse each subset $s_i$ we add the beginning index $b_i$ of $s_i$ to $B$ when the Jaccard distances $J_1 \geq 0.4$ or $J_2 \geq 0.4$ as given by Equation 5.

We define $J_j, j = \{1, 2\}$ by

$$J_j = J(s_i, V_j) = \frac{|s_i \cap V_j|}{|s_i| + |V_j| - |s_i \cap V_j|} \tag{5}$$

The Jaccard distances $J_1, J_2$ were chosen through a grid search from $[0.2, 0.8]$. The length of the masks $V_1, V_2$ were chosen through a grid search across the training data to minimize the number of false positives and maximize accuracy.

Once we have each beginning index $b_i$ of the range $s_i$, we merge nearby $b_i$s and use the middle of each set $b$ as an index of $T$ describing an indoor/outdoor transition. At the end of this step, $B$ contains all the IO transitions $b$ into and out of the building. The overall process is illustrated by Figure 2 and described in Algorithm 1.

## 4.4 ESTIMATING THE DEVICE'S VERTICAL HEIGHT

This part of the system determines the vertical offset measured in meters between the device's in-building location, and the device's last known IO transition. In previous work (Song et al., 2014) suggested the use of a reference barometer or beacons as a way to determine the entrances to a building. Our second key contribution is to use the LSTM IO predictions to help our system identify these indoor transitions into the building. The LSTM provides a self-contained estimator of a building's entrance without relying on external sensor information on a user's body or beacons placed inside a building's lobby.

This algorithm starts by identifying the last known transition into a building. This is relatively straightforward given the set of IO transitions $B$ produced by the previous step in the system (section 4.3). We can simply grab the last observation $b_n \in B$ and set the reference pressure $p_0$ to the lowest

device pressure value within a 15-second window around $b_n$. A 15-second window accounts for the observed 15-second lag that the GPS sensor needs to release the location lock from serving satellites.

The second datapoint we use in our estimate is the device's current pressure reading $p_1$. To generate the relative change in height $m_\Delta$ we can use the international pressure equation (6) (Milette & Stroud, 2012).

$$m_\Delta = f_{floor}(p_0, p_1) = 44330(1 - (\frac{p_1}{p_0})^{\frac{1}{5.255}}) \tag{6}$$

As a final output of this step in our system we have a scalar value $m_\Delta$ which represents the relative height displacement measured in meters, between the entrance of the building and the device's current location.

### 4.5 RESOLVING AN ABSOLUTE FLOOR LEVEL

This final step converts the $m_\Delta$ measurement from the previous step into an absolute floor level. This specific problem is ill-defined because of the variability in building numbering systems. Certain buildings may start counting floors at 1 while others at 0. Some buildings also avoid labeling the 13th floor or a maintenance floor. Heights between floors such as lobbies or food areas may be larger than the majority of the floors in the building. It is therefore tough to derive an absolute floor number consistently without prior knowledge of the building infrastructure. Instead, we predict a floor level indexed by the cluster number discovered by our system.

We expand on an idea explored by Xia et al. (2015) to generate a very accurate representation of floor heights between building floors through repeated building visits. The authors used clusters of barometric pressure measurements to account for drift between sensors. We generalize this concept to estimate the floor level of a device accurately. First, we define the distance between two floors within a building $d_{i,j}$ as the tape-measure distance from carpet to carpet between floor $i$ and floor $j$. Our first solution aggregates $m_\Delta$ estimates across various users and their visits to the building. As the number $M$ of $m_\Delta$'s increases, we approximate the true latent distribution of floor heights which we can estimate via the observed $m_\Delta$ measurement clusters $K$. We generate each cluster $k_i \in K$ by sorting all observed $m_\Delta$'s and grouping points that are within 1.5 meters of each other. We pick 1.5 because it is a value which was too low to be an actual $d_{i,j}$ distance as observed from an 1107 building dataset of New York City buildings from the Council on tall buildings and urban habitat (sky, 2017). During prediction time, we find the closest cluster $k$ to the device's $m_\Delta$ value and use the index of that cluster as the floor level. Although this actual number may not match the labeled number in the building, it provides the true architectural floor level which may be off by one depending on the counting system used in that building. Our results are surprisingly accurate and are described in-depth in section 5.

When data from other users is unavailable, we simply divide the $m_\Delta$ value by an estimator $\hat{m}$ from the sky (2017) dataset. Across the 1107 buildings, we found a bi-modal distribution corresponding to office and residential buildings. For residential buildings we let $\hat{m}_r = 3.24$ and $\hat{m}_o = 4.02$ for office buildings, Figure 6 shows the dataset distribution by building type. If we don't know the type of building, we use $\hat{m} = 3.63$ which is the mean of both estimates. We give a summary of the overall algorithm in the appendix (2).

## 5 EXPERIMENTS AND RESULTS

We separate our evaluation into two different tasks: The indoor-outdoor classification task and the floor level prediction task. In the indoor-outdoor detection task we compare six different models, LSTM, feedforward neural networks, logistic regression, SVM, HMM and Random Forests. In the floor level prediction task, we evaluate the full system.

### 5.1 INDOOR-OUTDOOR CLASSIFICATION RESULTS

In this first task, our goal is to predict whether a device is indoors or outdoors using data from the smartphone sensors.

Table 1: Model performance on validation and training set.

| Model | Validation Accuracy | Test Accuracy |
|-------|---------------------|---------------|
| LSTM | 0.923 | **0.903** |
| Feedforward | **0.954** | 0.903 |
| SVM | 0.956 | 0.876 |
| Random Forest | 0.974 | 0.845 |
| Logistic Regression | 0.921 | 0.676 |
| HMM | 0.976 | 0.631 |

All indoor-outdoor classifiers are trained and validated on data from 35 different trials for a total of 5082 data points. The data collection process is described in section 3.1. We used 80% training, 20% validation split. We don't test with this data but instead test from separately collected data obtained from the procedure in section 3.1.1.

We train the LSTM for $24$ epochs with a batch size of $128$ and optimize using Adam (Kingma & Ba, 2014) with a learning rate of $0.006$. We chose the learning-rate, number of layers, $d$ size, number of hidden units and dropout through random search (Bergstra & Bengio, 2012). We designed the LSTM network architecture through experimentation and picked the best architecture based on validation performance. Training logs were collected through the python test-tube library (Falcon, 2017) and are available in the GitHub repository.

**LSTM architecture:** Layers one and two have 50 neurons followed by a dropout layer set to $0.2$. Layer 3 has two neurons fed directly into a one-neuron feedforward layer with a sigmoid activation function.

Table 1 gives the performance for each classifier we tested. The LSTM and feedforward models outperform all other baselines in the test set.

## 5.2 FLOOR LEVEL PREDICTION RESULTS

Table 2: Floor level prediction error across 63 test trials. The left side of each column shows accuracy when floor to ceiling height $m = 4.02$. The right side shows accuracy when $m$ is conditioned on the building. Exact floor column shows the percent of the 63 trials which matched the targer floor exactly. The $\pm 1$ column is the percent of trials where the prediction was off by one.

| Classifier (m=4.02 \| m=bldg conditional) | Exact Floor | $\pm 1$ floor | $> \pm 1$ floor |
|-------------------------------------------|-------------|---------------|-----------------|
| LSTM | 0.65 \| 1.0 | 0.33 \| 0 | 0.016 \| 0 |
| Feedforward | 0.65 \| 1.0 | 0.33 \| 0 | 0.016 \| 0 |
| SVM | 0.65 \| 1.0 | 0.33 \| 0 | 0.016 \| 0 |
| Random Forest | 0.65 \| 1.0 | 0.33 \| 0 | 0.016 \| 0 |
| Logistic Regression | 0.65 \| 1.0 | 0.33 \| 0 | 0.016 \| 0 |
| HMM | 0.619 \| 0.984 | 0.365 \| 0 | 0.016 \| 0.015 |

We measure our performance in terms of the number of floors traveled. For each trial, the error between the target floor $f$ and the predicted floor $\hat{f}$ is their absolute difference. Our system does not report the absolute floor number as it may be different depending on where the user entered the building or how the floors are numbered within a building. We ran two tests with different $m$ values. In the first experiment, we used $m = m_r = 4.02$ across all buildings. This heuristic predicted the correct floor level with $65\%$ accuracy. In the second experiment, we used a specific $m$ value for each individual building.

This second experiment predicted the correct floor with $100\%$ accuracy. These results show that a proper $m$ value can increase the accuracy dramatically. Table 2 describes our results. In each trial, we either walked up or down the stairs or took the elevator to the destination floor, according to the

procedure outlined in section 3.1.1. The system had no prior information about the buildings in these trials and made predictions solely from the classifier and barometric pressure difference.

## 5.3 FLOOR LEVEL CLUSTERING RESULTS

Table 3: Estimated distances $d_{i,j}$ between floor $i$ and floor $j$ in the Uris Hall building.

| Floor range | Estimated $d_{i,j}$ | Actual $d_{i,j}$ |
|---|---|---|
| 1-2 | 5.17 | 5.46 |
| 2-3 | 3.5 | 3.66 |
| 3-4 | 3.4 | 3.66 |
| 4-5 | 3.45 | 3.5 |
| 5-6 | 3.38 | 3.5 |
| 6-7 | 3.5 | 3.5 |
| 7-8 | 3.47 | 3.5 |

In this section, we show results for estimating the floor level through our clustering system. The data collected here is described in detail in section 3.1.2. In this particular building, the first floor is 5 meters away from the ground, while the next two subsequent floors have a distance of 3.65 meters and remainder floors a distance of 3.5. To verify our estimates, we used a tape measure in the stairwell to measure the distance between floors from "carpet to carpet." Table 3 compares our estimates against the true measurements. Figure 5 in the appendix shows the resulting $k$ clusters from the trials in this experiment.

## 6 FUTURE DIRECTION

Separating the IO classification task from the floor prediction class allows the first part of our system to be adopted across different location problems. Our future work will focus on modeling the complete problem within the LSTM to generate floor level predictions from raw sensor readings as inspired by the works of Ghosh et al. (2016) and (Henderson et al., 2017).

## 7 CONCLUSION

In this paper we presented a system that predicted a device's floor level with $100\%$ accuracy in 63 trials across New York City. Unlike previous systems explored by Avinash et al. (2010), Song et al. (2014), Parviainen et al. (2008), Li et al. (2013), Xia et al. (2015), our system is completely self-contained and can generalize to various types of tall buildings which can exceed 19 or more stories. This makes our system realistic for real-world deployment with no infrastructure support required.

We also introduced an LSTM, that solves the indoor-outdoor classification problem with $90.3\%$ accuracy. The LSTM matched our baseline feedforward network, and outperformed SVMs, random forests, logistic regression and previous systems designed by Radu et al. (2014) and Zhou et al. (2012). The LSTM model also serves as a first step for future work modeling the overall system end-to-end within the LSTM.

Finally, we showed that we could learn the distribution of floor heights within a building by aggregating $m_\triangle$ measurements across different visits to the building. This method allows us to generate precise floor level estimates via unsupervised methods. Our overall system marries these various elements to make it a feasible approach to speed up real-world emergency rescues in cities with tall buildings.

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

## A APPENDIX

## B REAL-WORLD CONSIDERATIONS

In this section we explore potential pitfalls of our system in a real-world scenario and offer potential solutions.

### B.1 EXTERNAL PRESSURE CHANGES AND PRESSURE DRIFT

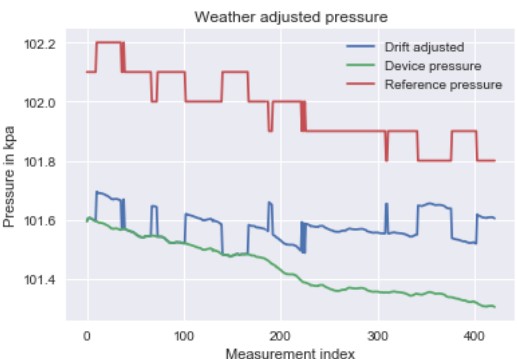

Figure 3: Adjusting device pressure from readings from a nearby station. The readings were mostly adjusted but the lack of resolution from the reference station made the estimate noisy throughout the experiment.

One of the main criticisms for barometric pressure based systems is the unpredictability of barometric pressure as a sensor measurement due to external factors and changing weather conditions. Critics have cited the discrepancy between pressure-sealed buildings and their environments, weather pattern changes, and changes in pressure due to fires (Roberson, 2014).

Li et al. (2013) used a reference weather station at a nearby airport to correct weather-induced pressure drift. They showed the ability to correct weather drift changes with a maximum error of 2.8 meters. Xia et al. (2015) also used a similar approach but instead adjust their estimates by reading temperature measurements obtained from a local weather station. We experimented with the method described by (Li et al., 2013) and conducted a single trial as a proof of concept. We measured the pressure reading $p$ from an iPhone device on a table over 7 hours while simultaneously querying weather data $w$ every minute. By applying the offset equation 7 we attempt to normalize the $p_i$ reading to the first $p_0$ reading generated by the device

$$p_0 \approx |w_i - w_0| + p_i \qquad (7)$$

we were able to stay close to the initial $p_0$ estimate over the experiment period. We did find that the resolution from the local weather station needed to be fine-grained enough to keep the error from drifting excessively. Figure 3 shows the result of our experiment.

### B.2 TIME SENSITIVITY

Our method works best when an offset $m_\Delta$ is predicted within a short window of making a transition within the building. (Li et al., 2013) explored the stability of pressure in the short term and found the pressure changed less than 0.1 hPa every 10 minutes (Li et al., 2013) on average. The primary challenge arises in the case when a user does not leave their home for an extended period of hours. In this situation, we can use the previously discussed weather offset method from section B.1, or via indoor navigation technology. We can use the device to list Wi-Fi access points within the building and tag each cluster location using RSSI fingerprinting techniques as described by Zhang et al. (2015) Ergen et al. (2014) and Tahat et al. (2016). With these tags in place, we can use the average

floor level tags of the nearest $n$ Wi-Fi access points once the delay between the building entrance and the last user location is substantial. We could not test this theory directly because of the limitations Apple places on their API to show nearby Wi-Fi access points to non-approved developers.

### B.3 DIFFERENCES BETWEEN BAROMETERS

Another potential source of error is the difference between barometric pressure device models. Xia et al. (2015) conducted a thorough comparison between seven different barometer models. They concluded that although there was significant variation between barometer accuracies, the errors were consistent and highly correlated with each device. They also specifically mentioned that the Bosch BMP180 barometer, the older generation model to the one used in our research, provided the most accurate measurements from the other barometers tested. In addition, Li et al. (2013) also conducted a thorough analysis using four different barometers. Their results are in line with Xia et al. (2015), and show a high correlation and a constant offset between models. They also noted that within the same model (Bosch BMP180) there was also a measurement variation but it was constant (Li et al., 2013).

### B.4 BATTERY IMPACT

Our system relies on continuous GPS and motion data collected on the mobile device. Continuously running the GPS and motion sensor on the background can have an adverse effect on battery life. Zhou et al. (2012) showed that GPS drained the battery roughly double as fast across three different devices. Although GPS and battery technology has improved dramatically since 2012, GPS still has a large adverse effect on battery life. This effect can vary across devices and software implementation. For instance, on iOS, the system has a dedicated chip that continuously reads device sensor data (Estes, 2013). This approach allows the system to stream motion events continuously without rapidly draining battery life. GPS data, however, does not have the same hardware optimization and is known to drain battery life rapidly. Nawarathne et al. (2014) conducted a thorough study of the impact of running GPS continuously on a mobile device. They propose a method based on adjusted sampling rates to decrease the negative impact of GPS on battery life. For real-world deployment, this approach would still yield fairly fine-grained resolution and would have to be tuned by a device manufacturer for their specific device model.

### B.5 Data Collection Procedure for Indoor-Outdoor Classifier Data

1) Start outside a building. 2) Turn Sensory on, set indoors to 0. 3) Start recording. 4) Walk into and out of buildings over the next $n$ seconds. 5) As soon as we enter the building (cross the outermost door) set indoors to 1. 6) As soon as we exit, set indoors to 0. 7) Stop recording. 8) Save data as CSV for analysis. This procedure can start either outside or inside a building without loss of generality.

### B.6 Data Collection Procedure for Indoor-Outdoor Classifier Data

1) Start outside a building. 2) Turn Sensory on, set indoors to 0. 3) Start recording. 4) Walk into and out of buildings over the next n seconds. 5) As soon as we enter the building (cross the outermost door) set indoors to 1. 6) Finally, enter a building and ascend/descend to any story. 7) Ascend through any method desired, stairs, elevator, escalator, etc. 8) Once at the floor, stop recording. 9) Save data as CSV for analysis.

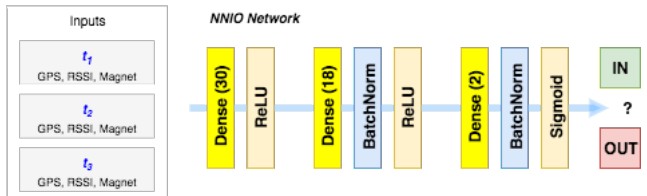

Figure 4: feedforward network architecture. A simple, three fully connected layer network. FC30 - FC18 - FC2. Inputs are sensor readings at times $t_1, t_2, t_3$.

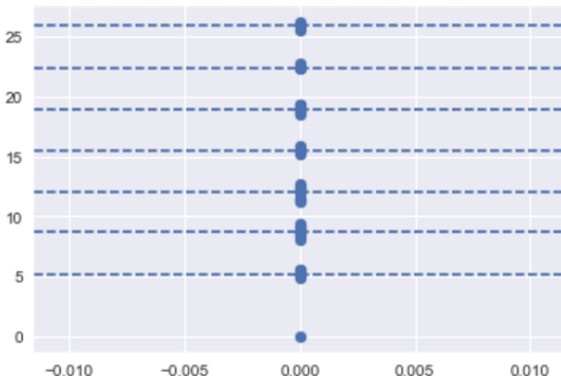

Figure 5: Distribution of $m_\Delta$ measurements across 41 trials in the Uris Hall building in New York City. A clear $d_{i,j}$ size difference is specially noticeable at the lobby. Each dotted line corresponds to an actual floor in the building learned from clustered data-points.

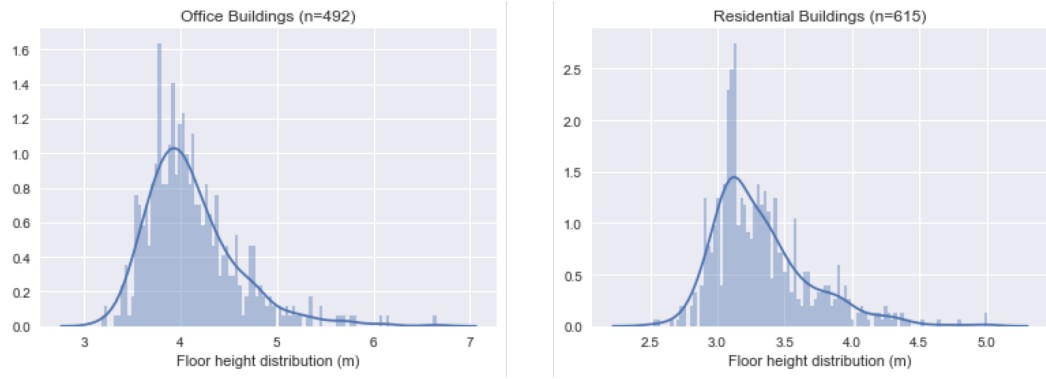

Figure 6: Distribution of "floor to floor" heights $d_{i,j}$ across 615 office (left) and 492 residential buildings (right) in New York City.

Table 4: Description of buildings used in our experiments. Material is the perceived material on the outside of the building.

| Bldg Name | Stories | Material | Trials |
|---|---|---|---|
| 10 Rockefeller Plz | 17 | Concrete | 10 |
| Uris Hall (CU) | 12 | Concrete | 14 |
| Mudd (CU) | 19 | Concrete | 10 |
| Noco (CU) | 14 | Glass | 10 |
| Social Work (CU) | 13 | Brick/Glass | 20 |

Table 5: Example data points collected for the indoor-outdoor classifier. Feature vector $X$ is constructed from 3 readings at consecutive time intervals $t_i$. $y$ is the middle point's IO label (1* in this case).

| $t_i$ | rssi | GV | GH | GS | M | P | IO (y) |
|---|---|---|---|---|---|---|---|
| 1 | -82 | 76.05 | 1414 | -1 | 1015.3 | 100.7 | 0 |
| 2 | -82 | 48 | 30 | 0.36 | 1019.4 | 100.3 | 1* |
| 3 | -82 | 48 | 30 | 0.36 | 1019.4 | 100.2 | 1 |

## C   ALGORITHM REFERENCES

---

**Algorithm 1** Find Indoor Outdoor transitions

---

1: **procedure** FINDIOINDEXES(T)
2:     $V_1 = [1, 1, 1, 1, 1, 0, 0, 0, 0, 0]$
3:     $V_2 = [0, 0, 0, 0, 0, 1, 1, 1, 1, 1]$
4:     $S = []$
5:                                                              ▷ Find Jacc $\geq 0.4$
6:     **for** $i \in \{1, ..., |T| - |V_1|\}$ **do**
7:         $s_i = \{t_i, ..., t_{i+|V_1|}\}$
8:         **if** $J(s_i, V_1) >= 0.4$ or $J(s_i, V_2) >= 0.4$ **then**
9:             $S \leftarrow i$
10:                                                             ▷ Merge $s_i$ ranges
11:     $merged = []$
12:     **for** each $b_i \in S$ **do**
13:         **if** $b_i + 2 <= b_{i-1}$ **then**
14:             $merged[i] \leftarrow \text{Merge}(b_i, b_{i-1})$
15:
16:     $transitions = []$
17:     **for** each $m_i \in merged$ **do**
18:         $transitions \leftarrow Middle(m_i)$
19: **return** $transitions$

---

**Algorithm 2** Predict Floor

---

1: **procedure** PREDICTFLOOR
2:     $T = CaptureData(GPS, baro, RSSI, magnet)$
                                                                ▷ Classify each reading as IO
3:     **for** $t \in T$ **do**
4:         $t_{IO} = NNIO(t)$
5:                                                              ▷ Locate last IO transition
6:     $B = FindIOTransitionIndexes(T)$
7:     $b_n = B[n]$
8:     **if** $b_{n,IO} = 0$ **then return** "outdoors"
9:                                                              ▷ Compute $m_\Delta$
10:     $p_1 = devicePressure(T_n)$
11:     $p_0 = devicePressure(b_n)$
12:     $m_\Delta = f_{floor}(p_0, p_1)$
13:                                                             ▷ Find the nearest $k$ cluster
14:     **for** $k_i \in K$ **do**
15:         $\hat{f} = |k_i - m_\Delta| \leq 1.5$
16:                                                             ▷ Estimate floor level
17:     **if** exists $\hat{f}$ **then**
18:         $\hat{f} = i$
19:     **else**
20:         $\hat{f} = \frac{m_\Delta}{bestM()}$
     **return** $\hat{f}$

---

