# OpenReview forum: "Predicting Floor-Level for 911 Calls with Neural Networks and Smartphone Sensor Data"
_ICLR.cc/2018/Conference — Accept (Poster)_

### Official Review · AnonReviewer2 · 2017-11-26
**A simple but useful method that serves a practical purpose well; improvements needed in writing and experimental comparisons.**

**Rating:** 7
**Confidence:** 5

**Review:**

The paper proposes a two-step method to determine which floor a mobile phone is on inside a tall building.
An LSTM RNN classifier analyzes the changes/fading in GPS signals to determine whether a user has entered a building.  Using the entrance point's barometer reading as a reference, the method calculates the relative floor the user has moved to using a well known relationship between heights and barometric readings.

The paper builds on a simple but useful idea and is able to develop it into a basic method for the goal.  The method has minimal dependence on prior knowledge and is thus expected to have wide applicability, and is found to be sufficiently successful on data collected from a real world context.  The authors present some additional explorations on the cases when the method may run into complications.

The paper could use some reorganization.   The ideas are presented often out of order and are repeated in cycles, with some critical details that are needed to understand the method revealed only in the later cycles.   Most importantly, it should be stated upfront that the outdoor-indoor transition is determined using the loss of GPS signals.  Instead, the paper elaborates much on the neural net model but delays until the middle of p.4 to state this critical fact. However once this fact is stated, it is obvious that the neural net model is not the only solution.

The RNN model for Indoor/Outdoor determination is compared to several baseline classifiers.  However these are not the right methods to compare to -- at least, it is not clear how you set up the vector input to these non-auto-regressive classifiers. You need to compare your model to a time series method that includes auto-regressive terms, or other state space methods like Markov models or HMMs.

Other questions:

p.2,  Which channel's RSSI is the one included in the data sample per second?

p.4,  k=3, what is k?

Do you assume that the entrance is always at the lowest floor? What about basements or higher floor entrances?  Also, you may continue to see good GPS signals in elevators that are mounted outside a building, and by the time they fade out,  you can be on any floor reached by those elevators.

How does each choice of your training parameters affect the performance?  e.g. number of epoches, batch size, learning rate.   What are the other architectures considered?  What did you learn about which architecture works and which does not?  Why?

As soon as you start to use clustering to help in floor estimation, you are exploiting prior knowledge about previous visits to the building.  This goes somewhat against the starting assumption and claim.

---

> ### Author Response · Authors · 2017-12-06
> **New results. We turned the problem into classification by fixing a window in the series of size (k=3). Updating paper with structural suggestions.**
>
> Thank you for your feedback! We're working on adding your suggestions and will post an update in the next few weeks.
>
> Wanted to let you know we've improved the results from 91% to 100% by adjusting our regularization mechanism in the LSTM. We'll make the appropriate changes to the paper.
>
> "The paper could use some reorganization"
> 1. Agreed and the updated draft will have:
>     - Cleaner organization
>     - Upfront clarification about the GPS signal
>     - Shortened discussion about the neural net model
>
> "The RNN model for Indoor/Outdoor determination is compared to several baseline classifiers."
> 2. The problem is reduced to classification by creating a fixed window of width k (in our case, k=3) where the middle point is what we're trying to classify as indoors/outdoors.
>     - Happy to add the HMM comparison.
>     - Happy to add a time series comparison.
>
> "p.2,  Which channel's RSSI is the one included in the data sample per second?
> "
> 3. We get the RSSI strength as proxied by the iPhone status bar. Unfortunately, the API to access the details of that signal is private. Therefore, we don't have that detailed information. However, happy to add clarification about how exactly we're getting that signal (also available in the sensory app code).
>
>
> 4. k is the window size. Will clarify this.
>
> "Do you assume that the entrance is always at the lowest floor? What about basements or higher floor entrances? "
>
> 5. We actually don't assume the entrance is on the lower floors. In fact, one of the buildings that we test in has entrances 4 stories appart. This is where the clustering method shines. As soon as the user enters the building through one of those lower entrances, the floor-level indexes will update because it will detect another cluster.
>
>
> "Also, you may continue to see good GPS signals in elevators that are mounted outside a building, and by the time they fade out,  you can be on any floor reached by those elevators."
> 6. Yup, this is true. Unfortunately this method does heavily rely on the indoor/outdoor classifier.
>     - We'll add a brief discussion to highlight this issue.
>
>
> "How does each choice of your training parameters affect the performance?  e.g. number of epoches, batch size, learning rate.   What are the other architectures considered?  What did you learn about which architecture works and which does not?  Why?
> "
> 7. We can add a more thorough description about this and provide training logs in the code that give visibility into the parameters for each experiment and the results.
>     - The window choice (k) actually might be the most critical hyperparameter (next to learning rate). The general pattern is that a longer window did not help much.
>     - The fully connected network actually does surprisingly well but the RNN generalizes slightly better. A 1-layer RNN did not provide much modeling power. It was the multi-layer model that added the needed complexity to capture these relationships. We also tried bi-directional but it failed to perform well.
>
> "As soon as you start to use clustering to help in floor estimation, you are exploiting prior knowledge about previous visits to the building.  This goes somewhat against the starting assumption and claim.
> "
> 8. Fair point. We provide a prior for each situation will will get you pretty close to the correct floor-level. However, it's impossible to get more accurate without building plans, beacons or some sort of learning. We consider the clustering method more of the learning approach: It updates the estimated floor heights as either the same user or other users walk in that building. In the case where the implementer of the system (ie. A company), only wants to use a single-user's information and keep it 100% on their device, the clustering system will still work using that user's repeated visits. In the case where a central database might aggregate this data, the clusters for each building will develop a lot faster and converge on the true distribution of floor heights in a buillding.

---

> ### Author Response · Authors · 2018-01-05
> **HMM added**
>
> Hello. We've added the HMM baseline. We apologize for the delay, we wanted to make sure we set the HMM baseline as rigorous as possible.
>
> The code is also available for your review.
>
> Thank you once again for your feedback!

---

### Official Review · AnonReviewer1 · 2017-11-27
**The paper combines existing methods to outperform baseline methods on floor level estimation. Limitations of their approach are not explored.**

**Rating:** 6
**Confidence:** 3

**Review:**

The authors motivate the problem of floor level estimation and tackle it with a RNN. The results are good. The models the authors compare to are well chosen. As the paper foremost provides application (and combination) of existing methods it would be benefitial to know something about the limitations of their approach and about the observed prequesits.

---

> ### Author Response · Authors · 2017-12-05
> **Appendix A, section B has potential pitfalls**
>
> Thank you for your valuable feedback!
> In Appendix A, section B we provide a lengthy discussion about potential pitfalls of our system in a real-world scenario and offer potential solutions.
>
> Was there something in addition to this that you'd like to see?

---

> ### Author Response · Authors · 2018-01-05
> **Code released, results updated**
>
> Dear reviewer,
>
> We've released a main update listed above. Please let us know if there's anything we can help clarify!
>
> Thank you once again for your feedback!

---

### Official Review · AnonReviewer3 · 2017-11-27
**A fairly simple application of existing methods to a problem, and there remain some methodological issues**

**Rating:** 6
**Confidence:** 4

**Review:**

Update: Based on the discussions and the revisions, I have improved my rating. However I still feel like the novelty is somewhat limited, hence the recommendation.

======================

The paper introduces a system to estimate a floor-level via their mobile device's sensor data using an LSTM to determine when a smartphone enters or exits a building, then using the change in barometric pressure from the entrance of the building to indoor location. Overall the methodology is a fairly simple application of existing methods to a problem, and  there remain some methodological issues (see below).

General Comments
- The claim that the bmp280 device is in most smartphones today doesn’t seem to be backed up by the “comScore” reference (a simple ranking of manufacturers).  Please provide the original source for this information.
- Almost all exciting results based on RNNs are achieved with LSTMs, so calling an RNN with LSTM hidden units a new name IOLSTM seems rather strange - this is simply an LSTM.
- There exist models for modelling multiple levels of abstraction, such as the contextual LSTM of [1]. This would be much more satisfying that the two level approach taken here, would likely perform better, would replace the need for the clustering method, and would solve issues such as the user being on the roof.  The only caveat is that it may require an encoding of the building (through a one-hot encoding) to ensure that the relationship between the floor height and barometric pressure is learnt. For unseen buildings a background class could be used, the estimators as used before, or aggregation of the other buildings by turning the whole vector on.
- It’s not clear if a bias of 1 was added to the forget gate of the LSTM or not. This has been shown to improve results [2].
- Overall the whole pipeline feels very ad-hoc, with many hand-tuned parameters.  Notwithstanding the network architecture, here I’m referring to the window for the barometric pressure, the Jaccard distance threshold, the binary mask lengths, and the time window for selecting p0.
- Are there plans to release the data and/or the code for the experiments? Currently the results would be impossible to reproduce.
- The typo of accuracy given by the authors is somewhat worrying, given that the result is repeated several times in the paper.

Typographical Issues
- Page 1:  ”floor-level accuracy” back ticks
- Page  4:   Figure  4.1→Figure  1;  Nawarathne  et  al  Nawarathne  et  al.→Nawarathne et al.
- Page 6:  ”carpet to carpet” back ticks
- Table 2:  What does -4+ mean?
- References.  The references should have capitalisation where appropriate.For example,  Iodetector→IODetector,  wi-fi→Wi-Fi,  apple→Apple, iphone→iPhone, i→I etc.

[1]  Shalini Ghosh, Oriol Vinyals, Brian Strope, Scott Roy, Tom Dean, and LarryHeck. Contextual LSTM (CLSTM) models for large scale NLP tasks. arXivpreprint arXiv:1602.06291, 2016.
[2]  Rafal Jozefowicz, Wojciech Zaremba, and Ilya Sutskever.  An empirical exploration of recurrent network architectures.  InProceedings of the 32nd International Conference on Machine Learning (ICML-15), pages 2342–2350,2015

---

> ### Author Response · Authors · 2017-12-05
> **New model accuracy is 100% with no margin of error. Added device references, discussion about new model, and code + data can be public if requested beforehand**
>
>
> Thank you so much for your valuable feedback! I want to preface the breakdown below by letting you know that we added time-distributed dropout which helped our model's accuracy. The new accuracy is 100% with no margin of error in the floor number.
>
> 1. As of June 2017 the market share of phones in the US is 44.9% Apple and 29.1% Samsung [1]. 74% are iPhone 6 or newer [2]. The iPhone 6 has a barometer [3]. Models after the 6 still continue to have a barometer.
> For the Samsung phones, the Galaxy s5 is the most popular [4], and has a barometer [5].
>
>
> [1] https://www.prnewswire.com/news-releases/comscore-reports-june-2017-us-smartphone-subscriber-market-share-300498296.html
> [2] https://s3.amazonaws.com/open-source-william-falcon/911/2017_US_Cross_Platform_Future_in_Focus.pdf
> [3] https://support.apple.com/kb/sp705?locale=en_US
> [4] https://deviceatlas.com/blog/most-popular-smartphones-2016
> [5] https://news.samsung.com/global/10-sensors-of-galaxy-s5-heart-rate-finger-scanner-and-more
>
> 2. Makes sense, we separated it for the non deep learning audience trying to understand it. However, happy to update everything to say LSTM.
> 3. Thanks for this great suggestion. We had experimented with end-to-end models but decided against it.  We did have a seq2seq model that attempted to turn the sequence of readings into a sequence of meter offsets. It did not fully work, but we're still experimenting with it. This model does not however get rid of the clustering step.
>
> An additional benefit of separating this step from the rest of the model is that it can be used as a stand-alone indoor/outdoor classifier.
>
> I'll address your concerns one at a time:
>     a. In which task would it perform better? The indoor-outdoor classification task or the floor prediction task?
>     c. What about this model would solve the issue of the user being on the roof?
>     d. Just to make sure I understand, the one-hot encoding suggestion aims to learn a mapping between the floor height and the barometric pressure which in turn removes the need for clustering?
>     e. This sounds like an interesting approach, but seems to fall outside of the constraint of having a self-contained model which did not need prior knowledge. Generating a one-hot encoding for every building in the world without a central repository of building plans makes this intractable.
>
> 4. We used the bias (tensorflow LSTM cell). https://www.tensorflow.org/api_docs/python/tf/contrib/rnn/LSTMCell
> 5. Happy to add explanations for why the "ad-hoc" parameters were chosen:
>     a. Jaccard window, binary mask lengths, and window length were chosen via grid search.
>     b. Will add those details to the paper.
>
> 6. Yes! All the data + code will be made public after reviews. However, if you feel strongly about having it before, we can make it available sooner through an anonymous repository. In addition, we're planning on releasing a basic iOS app which you'll be able to download from the app store to run the model on your phone and see how it works on any arbitrary building for you.
>
> 7. Yes, many typos. Apologize for that. We did a last minute typo review too close to the deadline and missed those issues. This is in fact going to change now that we've increased the model accuracy to 100% with no floor margin of error.
>
> We're updating the paper now and will submit a revised version in the coming weeks

---

### Comment · AnonReviewer3 · 2018-01-03
**Drop in accuracy (?) and baselines**

Can you explain why in the table 1 in the revision from 29th October the validation and test accuracy of the LSTM are 0.949 and 0.911 and in the most recent version they have dropped to 0.935 and 0.898 (worse than the baselines)?

Also I agree with the statement by reviewer 2:

"The RNN model for Indoor/Outdoor determination is compared to several baseline classifiers.  However these are not the right methods to compare to -- at least, it is not clear how you set up the vector input to these non-auto-regressive classifiers. You need to compare your model to a time series method that includes auto-regressive terms, or other state space methods like Markov models or HMMs."

It seems like no changes have been made to address this.

---

> ### Author Response · Authors · 2018-01-03
> **Updating HMM results today**
>
> Hi. Once again, thank you for your feedback.
>
> re HMM:
> Yes, currently finalizing the HMM baseline right now and will be adding to the paper by today.
>
> re Model accuracy:
> These are the results from the latest hyperparameter optimization. We're verifying these today once the tests complete. Although the model accuracy dropped for the indoor/outdoor classification task, it increased to 100% with no margin of error for the floor prediction task. The other baselines don't achieve near the same result on the floor prediction task. However, we're running a final optimization today to ensure we have the best results given the hyperparameter search.
>
> We can add those model results to the floor prediction task for clarification.

---

> ### Author Response · Authors · 2018-01-05
> **HMM added, code released**
>
> Hello. We've clarified these issues in the primary post above. Please let us know if we've addressed your concerns.
>
> Thank you once again for your valuable feedback

---

### Author Response · Authors · 2018-01-05
**HMM added, code released, models verified again**

Thank you once again for your feedback during these last few weeks. We've gone ahead and completed the following:
1. Added the HMM baseline
2. Reran all the models and updated the results. The LSTM and feedforward model performed the same on the test set. We've reworded the results and method page to reflect this.
3. By increasing the classifier accuracy we improved the floor-prediction task to 100% with no margin of error on the floor predictions.
4. We tried the hierarchical LSTM approach as suggested but did not get a model to work in the few weeks we experimented with it. It looks promising, but it'll need more experimentation. We included this approach in future works section.
5. We released all the code at this repository: https://github.com/blindpaper01/paper_FMhXSlwRYpUtuchTv/

Although the code is mostly organized, works and is commented it'll be polished up once it needs to be released to the broader community. The Sensory app was not released yet to preserve anonymity.

6. Fixed typos (some of the numbering typos are actually from the ICLR auto-formatting file).

Please let us know if there's anything else you'd like us to clarify.
Thank you so much for your feedback once again!

---

> ### Comment · AnonReviewer3 · 2018-01-05
> **Floor prediction results**
>
> For the floor prediction task the result given is using the LSTM right (I don't think it's actually specified)? Do you have results for the baselines for this?

---

> > ### Author Response · Authors · 2018-01-05
> > **Floor prediction results**
> >
> > Yes, LSTM only. Generating others now

---

> > ### Author Response · Authors · 2018-01-05
> > **Floor prediction results**
> >
> > The table has been updated. The algorithm did fairly well on this task when using each of the classifiers with the exception of the HMM. The difference between classifiers on this task would likely come through when the possibility of acquiring a GPS lock during a trial comes up such as a glass elevator on the outside of the building. In this case, the LSTM would likely produce less false positives as indicated by the increased accuracy in the IO classification task. Fewer false-positives mean that the algorithm would likely identify the correct anchor barometric pressure point at the entrance to the building instead of a stairwell or entering the building from the glass elevator.

---

> > > ### Comment · AnonReviewer3 · 2018-01-05
> > > **Floor prediction results**
> > >
> > > Thanks for adding this. It starts to look like there's nothing to choose between the LSTM and Feedforward network ...

---

> > > > ### Author Response · Authors · 2018-01-05
> > > > **Floor prediction results**
> > > >
> > > > Based on this data, and these results, the line between both models is certainly more blurry. What is clear is that the neural network models do outperform the other models. We've changed some of the wording to highlight this point and not make it a strictly LSTM vs others approach but instead a neural network vs others approach. Although given more complicated examples, we think the LSTM would perform better on the IO task. However, generating more data is very time-consuming so it makes the overall problem difficult to model.
> > > >
> > > > But we believe the point you mentioned with hierarchical LSTMs is extremely relevant in this context because it allows a foundation on which to build future work to model the full problem end-to-end with a model based on LSTM architectures and the hierarchical approaches mentioned. We added this point in the future direction and certainly think it's feasible but it'll likely require more data and model design.
> > > >
> > > >  However, we're still fine-tuning the LSTM to see why there was a 1% drop.

---

### Decision · Program_Chairs · 2018-01-29
**ICLR 2018 Conference Acceptance Decision**

**Decision:**

Accept (Poster)

**Comment:**

Reviewers agree that the paper is well done and addresses an interesting problem, but uses fairly standard ML techniques.
The authors have responded to rebuttals with careful revisions, and improved results.

---

> ### Author Response · Authors · 2018-01-30
> **Thank you**
>
> Thanks to all the reviewers for the valuable feedback!